# Corrosion Behavior of Hydrotalcite Film on AZ31 Alloy in Simulated Body Fluid

**Jun Chen [1,2,3,*], Kai Kang [1], Yingwei Song [2], En-Hou Han [2], Sude Ma [1] and Jinqing Ao [1,3]**

[1]  Key Laboratory of Fluid and Power Machinery of Ministry of Education, School of Materials Science and Engineering, Xihua University, Chengdu 610039, China; 18328816957@163.com (K.K.); masude2007@163.com (S.M.); jinqingaowust@163.com (J.A.)

[2]  National Engineering Center for Corrosion Control, Institute of Metal Research, Chinese Academy of Sciences, 62 Wencui Road, Shenyang 110016, China; ywsong@imr.ac.cn (Y.S.); ehhan@imr.ac.cn (E.-H.H.)

[3]  Jiangsu Steel Tube Co., Ltd., Wuxi 214121, China

*   Correspondence: chenjun_812@163.com; Tel.: +86-28-87729250

**Abstract:** The hydrotalcite (HT) film is a promising bioactive coating for magnesium alloys. In the present study, we investigate the corrosion behavior of HT film in the simulated body fluid (SBF), and compare with which in NaCl solution. The HT film can provide a very plummy initial protection to the AZ31 alloy in SBF. The corrosion behavior of the HT film in the two solutions is quite different. When in 0.1 mol·L$^{-1}$ NaCl solution, the film is dissolved gradually, and filiform corrosion is predominant after 3 days immersion. While in Hank's solution, the thickness and composition of the film are changed. A corrosion products layer mainly consisted of Mg/Ca–PO$_4$$^{3-}$/HPO$_4$$^{2-}$, and minor of CaCO$_3$ is deposited on the top of HT film, which enhances the barrier effect of the HT film. As a result, except for local pit corrosion at several active places, most of the area of the coated sample still remains integral even after immersion for 15 days. It is demonstrated that the HT film has better corrosion protection effect in SBF than in NaCl solution.

**Keywords:** magnesium alloy; hydrotalcite film; simulated body fluid; corrosion

## 1. Introduction

Magnesium (Mg) and its alloys have been intensively studied as biodegradable implant materials since they possess many advantages as follows. (1) Good biocompatibility: Mg is an element essential to the human body with half of the total physiological Mg stored in bone tissue [1–3]. The in vitro result reported by Cheng et al. indicated that Mg does not present any cytotoxic effects on L929 and ECV304 cells [4]. The in vivo study also showed that Mg/Mg alloys are biocompatible [5]. (2) Good mechanical properties: Mg alloys have similar strength and elastic modulus with natural bones, and have the potential to minimize stress shielding [6,7]. (3) Close density with bones: Densities of Mg based metals (1.7–2.0 g cm$^{-3}$) are close to natural bones (1.8–2.1 g cm$^{-3}$) [8]. (4) Additionally, due to their unique biodegradable property, Mg based implants do not demand second surgery compared with those stainless steel and titanium based implants [9–11]. Both mechanical properties and degradation profiles should be required for the next-generation biodegradable stents [12]. However, Mg alloys undergo rapid corrosion under physiological conditions [13,14], resulting in an early loss of mechanical stability of the implant before the end of the healing process [15]. To solve the corrosion problem, the main method is to develop corrosion resistant alloys. Surface treatment is another commonly used method to slow down the corrosion rate of the Mg alloys.

Concerning the clinical application of biodegradable Mg implants, an ideal coating should be biocompatible and dissolvable in human body. From this point of view, hydrotalcite-like compounds

(HTs) film is a potential option. On the one hand, HTs have good biocompatibility and low toxicity. HTs have been approved for human clinical uses including drug release, as well as protein, nucleotides and DNA carriers [16–22]. For example, Riaz et al. pointed out that the drug-HT hybrids were quite instrumental because of their application as advanced anti-cancer drug delivery systems [19]. Data obtained by Perioli et al. suggested that HT was a suitable material able to improve the biopharmaceutical properties of class IV BCS drugs [20]. From the in vivo testing of adult male Sprague Dawley rats, Kwak et al. [18] concluded that the HT particles had little systemic effect at doses $\leq$200 mg kg$^{-1}$. On the other hand, HTs films have been applied to enhance the corrosion resistance of Mg alloys [23–28]. Lin and Chen found that the HT conversion films adhere well to the substrates, and provide a good protection to the Mg alloys in NaCl solution [24,26]. Furthermore, it is found that HTs solids are degradable when they are exposed to aqueous solutions [27,29]. Although, we have investigated the corrosion behavior of the HT film in NaCl solution, the protective property of this HT film to Mg in the physiological environment is not well understood. Hence, the aim of the present work is to investigate the corrosion resistance of the Mg-Al HT film as biologic coating in simulated physiological conditions. In addition, a comparison of the corrosion behavior of the HT film in NaCl solution and simulated body fluid (SBF) is also provided in this paper.

## 2. Materials and Methods

### 2.1. Fabrication of the Films

The material used in this study was extruded AZ31 alloy. The grain size of the alloy was not very even, the average value was about 16 μm. The surface of the samples was ground to 2000 grit SiC paper, ultrasonically cleaned in ethyl alcohol, and then dried in the cold air. The HT film was prepared by a two-step in situ growth method, and followed the same procedure according to the literature previously described [26]. Carbonic acid solution was prepared by bubbling $CO_2$ gas through 200 mL of distilled water for about 10 min at room temperature (20 $\pm$ 2 °C). Al containing solution was prepared by dissolving a pure Al panel into the 0.5 mol·L$^{-1}$ $Na_2CO_3$ solution at 60 °C. Then this Al containing solution was dropped into the carbonic acid solution until achieving the pH value of about 8 to form the pretreatment solution at 60 °C. The post treatment solution was based on the pretreatment solution using NaOH solution to adjust the pH value to 10.5, then heated to 80 °C. Samples were first immersed in the pretreatment solution with a continuous bubbling of $CO_2$ for 30 min and then immersed in the post treatment solution for 1.5 h to obtain the HT film.

### 2.2. Characterization

The morphology of the films was observed using an environmental scanning electronic microscope (ESEM, Philips XL30, FEI, OR, USA) equipped with an energy dispersive X-ray spectroscopy (EDS, X-act, Oxford Instruments, Oxford, UK). The cross section samples were first mounted using epoxy resin. Then they were ground to 5000 grit SiC paper. After that, they were polished with alumina powders (0.5 μm). Subsequently, samples were cleaned with ethyl alcohol, and dried in the cold air. The chemical composition was analyzed by EDS and X-ray photoelectron spectroscopy (XPS, ESCALAB 250, Thermo Fisher Scientific, Waltham, MA, USA). The XPS was probed using Al K$\alpha$ radiation (1486.6 eV). The power was 150 W, the pass energy was 50.0 eV and the step size was 0.1 eV. All energy values were corrected according to the adventitious C 1$s$ signal set at 284.6 eV. The data were analyzed with Xpspeak 4.1 software (v4.1, developed by Raymund Kwok, Hongkong, China).

Electrochemical tests were carried out using a ParStat 4000 potentiostat (Ametec, Berwyn, PA, USA). A classical three-electrode system was applied. The samples, a saturated calomel electrode (SCE) and a platinum plate, were used as working electrode, reference electrode and auxiliary electrode, respectively. The open circuit potential (OCP) of the AZ31 alloy with and without film was investigated by a potential vs. time curve with a sampling frequency of 100 s point$^{-1}$. The polarisation curves were obtained on an exposed area of 1 cm$^2$ at a constant voltage scan rate of 0.5 mV s$^{-1}$ after an initial delay

of 300 s. Immersion test was performed according to the GB 10124-88 of China [30]. The size of the samples for immersion tests was $25 \times 50 \times 2$ mm$^3$. The electrochemical and immersion tests were conducted in Hank's solution (NaCl 8.0 g/L, KCl 0.4 g/L, CaCl$_2$ 0.14 g/L, NaHCO$_3$ 0.35 g/L, C$_6$H$_6$O$_6$ 1.0 g/L, MgCl$_2$·6H$_2$O 0.1 g/L, MgSO$_4$·7H$_2$O 0.06 g/L, KH$_2$PO$_4$ 0.06 g/L, Na$_2$HPO$_4$·12H$_2$O 0.06 g/L) at $37 \pm 2$ °C. The corrosion products were removed in a chromic acid bath consisting of 180 g L$^{-1}$ CrO$_3$.

## 3. Results and Discussion

### 3.1. Morphology of the HT Film

Figure 1 shows the morphology of the coated sample. According to our previous work [26], the chemical composition of the film was mainly Mg$_6$Al$_2$(OH)$_{16}$CO$_3$·4H$_2$O. The optical photo in Figure 1a shows that the HT film is very homogeneous. Figure 1b presents a compact and smooth film, and micro-cracks hardly can be seen. Higher magnification morphology in Figure 1c clearly demonstrates a typical HT structure. The surface of the Mg substrate is completely covered with the dense and uniform blade-like flakes. From the cross-sectional image of the coated sample in Figure 1d, it is observed that the HT film is strongly adhered to the substrate, and there is no crack on the film, which indicates that the film is very compact. The thickness of the film is about 1.04 μm.

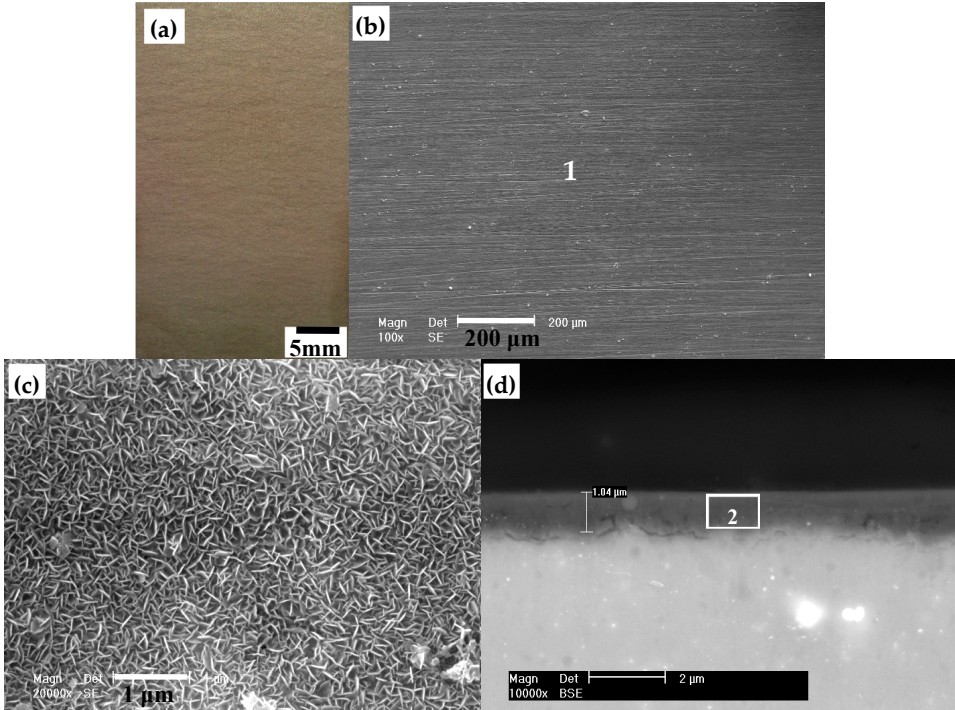

**Figure 1.** Morphology of the hydrotalcite film coated sample: (**a**) Optical photo; (**b**) surface morphology in low magnification; (**c**) surface morphology in high magnification; (**d**) cross-sectional morphology. (1 and 2 represent the EDS full scan region in Figure 1b and EDS testing region in Figure 1d, respectively. They correspond to the numbers in Table 1).

### 3.2. Electrochemical Corrosion Test

The potential vs. time curves of AZ31 alloy with and without film in Hank's solution is shown in Figure 2a. In the case of AZ31 substrate, the OCP keeps increasing from $-1.84$ to $-1.57$ V (vs. SCE) in the initial $1.66 \times 10^4$ s, and then drops down slightly. It implies that a corrosion products film is formed continuously on the alloy surface, and then the film breaks down. Afterwards, OCP fluctuated in a small range, indicating that the rupture and formation of the corrosion products film reach a dynamic equilibrium. The optical morphology of the AZ31 substrate after being soaked for $9.0 \times 10^4$ s

is also provided in the low section of Figure 2a. It is observed that there is a thin but non-homogeneous film on the surface, where many white-bright spots distribute randomly, indicating the rupture of the corrosion products film. In contrast, the performance of the sample coated with HT film is different. OCP of the coated sample increases from $-1.90$ to $-1.63$ V (vs. SCE) within $1.20 \times 10^4$ s. Subsequently, it keeps elevating with a relatively lower slope. It can be implied that the reaction between HT film and Hank's solution sustains during the whole immersion to form a corrosion products layer on the top, which is hardly stabilized. To the end of the testing, the OCP of the coated sample achieves to $-1.49$ V (vs. SCE). It is noteworthy that OCP fluctuates several times at the range around $1.20 \times 10^4$, $5.70 \times 10^4$ and $1.84 \times 10^5$ s, respectively. It may be because that local superficial pit corrosion occurs at some active spots, but these active areas are covered by corrosion products immediately. No corrosion pits are observed by naked eye on the surface of the coated sample after being soaked for $2.62 \times 10^5$ s (more than 3 days). It can be implied that the film can effectively delay the initial appearance of megascopic corrosion pit. Figure 2b shows the polarization curves of the AZ31 alloy with and without film in Hank's solution. The HT film slows down the corrosion rate of bare alloy by inhibiting both the cathodic hydrogen evolution and anodic dissolution reactions. The hydrogen evolution rate in the cathodic side is decreased. The anodic sides of the two curves are quite different. In curve 2, there is a passive tendency region. The breakdown potential ($E_b$) of the film coated sample is about $-1.34$ V, which is more positive than that of the substrate ($-1.51$ V). The corrosion current density ($i_{corr}$) of the substrate and the coated sample is approximately 23.64 and 5.54 $\mu A\ cm^{-2}$, respectively. The positive shift of the corrosion potential and the decrease of current density indicate that the HT film can improve the corrosion resistance of the AZ31 alloy in SBF.

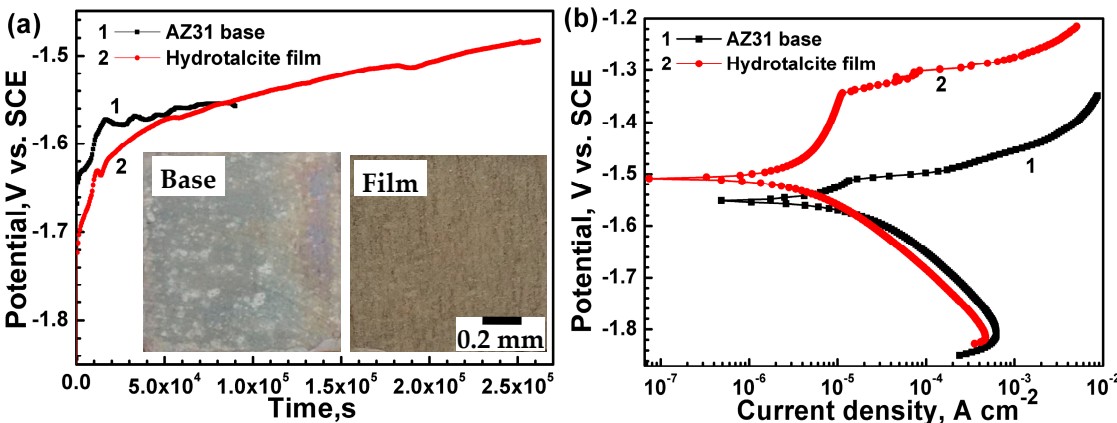

**Figure 2.** (**a**) Potential vs. time curves; (**b**) polarization curves of the AZ31 alloy with and without the hydrotalcite film in Hank's solution.

### 3.3. Corrosion Morphology of the HT Film

Optical images of the AZ31 alloy with and without HT film immersed in Hank's solution for different time are shown in Figure 3. After 3 days immersion, filiform corrosion has happened to more than half area of the bare alloy, which is corroded severely. The majority of the coated sample is not attacked after 15 days immersion. The immersion test also indicates that the HT film can offer good protection to the AZ31 alloy in SBF. However, compared to the morphology of the original HT film before immersion which is very homogeneous and smooth without any tubercle (Figure 1a), there are some white particles on the surface of the HT film after immersion test.

Figure 4 presents the surface morphology of the AZ31 alloy with HT film after immersion test for 15 days in Hank's solution. It is revealed from the low magnification morphology observation of the SEM image that lots of white corrosion products are deposited on the pit. However, these particles do not stack densely. It can be seen from the high magnification morphology that the microstructure of the top film changes greatly after immersion test. The baculiform particles stack tightly, which is quite different from the original HT microstructure before immersion (Figure 1c) than the curved

hexagonal platelets lying perpendicular to the substrate surface. After removing corrosion products, it is observed that the depth of pit is inhomogeneous, where it is deeper at the intermediate site (Figure 4c). It indicates that both vertical development and horizontal development of the pit corrosion occurred. Many shallow corrosion traces are also observed on the surface of the substrate.

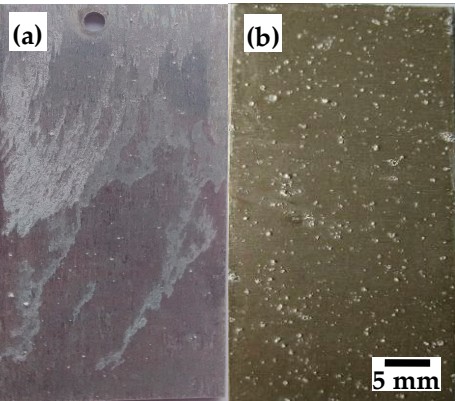

**Figure 3.** Optical corrosion morphology of: (**a**) The bare AZ31 alloy after immersion test for 3 days; (**b**) the hydrotalcite film coated sample after immersion tests for 15 days in Hank's solution.

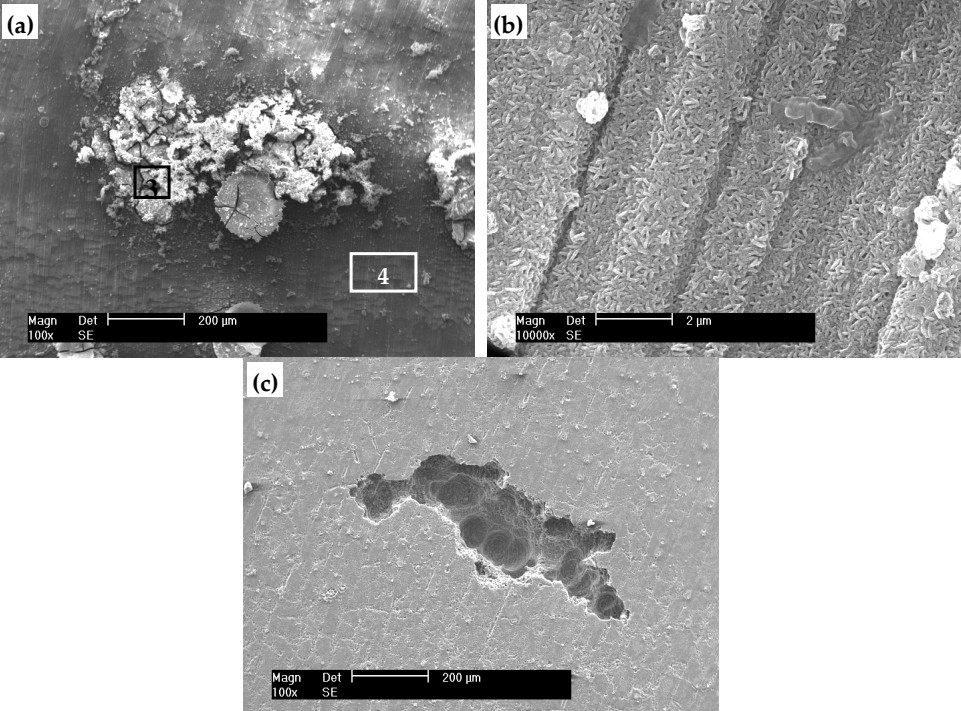

**Figure 4.** Surface morphology of the hydrotalcite film coated sample after immersion test for 15 days in Hank's solution: (**a**) The white particle area in low magnification; (**b**) the intact area in high magnification; (**c**) the pit area after removing corrosion products. (3 and 4 in Figure 4a represent the EDS testing regions and correspond to the numbers in Table 1.)

Figure 5 shows the cross-sectional morphology of the HT film coated sample after immersion test for 15 days in Hank's solution. Figure 5a displays a deep pit, the depth of which is about 126 μm, and it is filled up by corrosion products. In addition, there are many micro-cracks in the products block mass. In the high magnification image, there are three layers above the substrate, just like a "sandwich". The top layer is the corrosion products film covered above the HT film, the thickness of which is about 0.73 μm. The second layer is the HT film after immersion, but compared to the original HT film, the thickness of which is increased significantly (about 2.12 μm). There is an open crack, penetrating the

two layers directly to the substrate. Those micro-cracks can provide channels for the corrosion medium passing through to the substrate. As a result, the Mg substrate under the film is corroded, and another loose and discontinuous corrosion products film layer is formed underneath the HT film. In addition, there is a gap between the HT film and the bottom corrosion products layer. It may be caused by the mechanical ground because the corrosion products of the substrate underneath the HT film are loose and brittle.

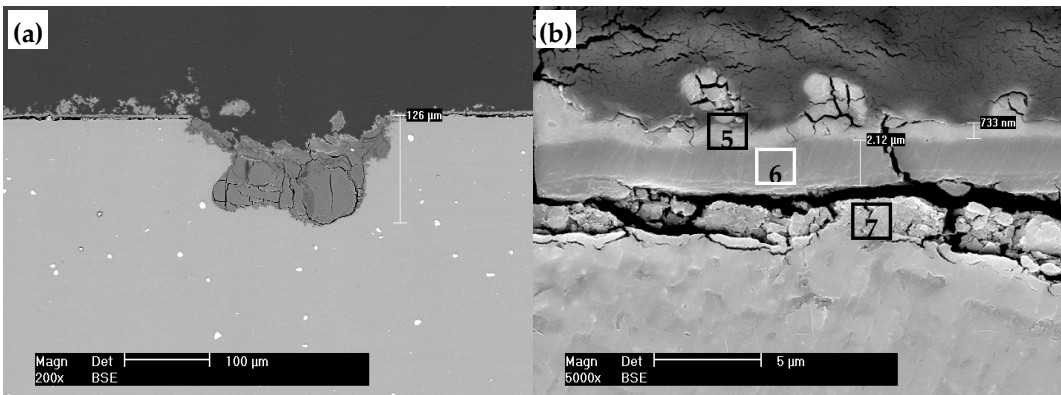

**Figure 5.** Cross-sectional morphology of the hydrotalcite film coated sample after immersion test for 15 days in Hank's solution: (**a**) Low magnification; (**b**) high magnification. (5, 6 and 7 in Figure 5b represent the EDS testing regions and correspond to the numbers in Table 1.)

### 3.4. Composition Analysis of the Film

The chemical composition of the film before and after immersion was analyzed by EDS and XPS. The content of various elements in different regions of the films in Figures 1, 4a and 5b marked by 1–7 is listed in Table 1. The original HT film only contains C, Mg, Al and O, and the content of Mg is very high (Region 1), indicating that the signal from the matrix is strong. While the composition of the film after immersion test is more complex, including C, O, Mg, Al, P and Ca, and the signal from the matrix Mg is weak (Region 4), implying that the detected information is mainly attributed to the film. It is because the thickness of the film after immersion is increased, and the result is in accordance with the cross-sectional morphology (compared Figures 1d and 5b). Furthermore, the atomic ratio of Mg and Al elements in Region 6 is decreased to 3.13:1, compared with Region 2 which is about 3.68:1. The main information from the composition comparison of the HT film before and after immersion is that Ca and P have been inserted into the HT structure. The corrosion products deposited on pit are composed of O, C, Mg, P and Ca, and the content of Ca is especially high (Region 3). According to the EDS analysis, the contents of Ca and P in the corrosion products film layer are high, but the Al content is very small (Region 5). The composition of the bottom corrosion products layer is C, O, Al, Mg, Ca and P (Region 7), and Mg is the main metal element. In addition, the content of Ca and P in the corrosion products film layer below the HT film is smaller than that in the top layer. It may be because that only small amount of electrolyte passed through into the HT film with poor liquidity.

**Table 1.** The content of various elements in different regions of the films in Figures 1, 4a and 5b marked by 1–7, respectively.

| Element (at. %) | C | O | Mg | Al | P | Ca |
|---|---|---|---|---|---|---|
| 1 | 15.67 | 21.00 | 59.54 | 03.80 | — | — |
| 2 | 16.53 | 42.96 | 31.85 | 8.66 | — | — |
| 3 | 19.36 | 42.94 | 04.02 | — | 12.57 | 21.11 |
| 4 | 20.09 | 48.88 | 16.91 | 05.19 | 04.99 | 03.94 |
| 5 | 37.12 | 40.36 | 06.18 | 00.62 | 09.04 | 06.67 |
| 6 | 21.84 | 48.57 | 17.17 | 05.49 | 04.65 | 02.28 |
| 7 | 25.35 | 40.90 | 23.78 | 02.60 | 05.85 | 01.52 |

The XPS analysis of the HT film after immersion test for 15 days in Hank's solution is shown in Figure 6. The C1s spectrum has two peaks. The strong peak is due to the adventitious hydrocarbons from the environment. Small shoulder at approximate 289.6 eV indicates the presence of $CaCO_3$. Figure 6b presents the high resolution XPS spectrum of O 1*s*, which is deconvoluted into three peaks. The peak at 533.7 eV can be attributed to P–OH [31,32]. The binding energy of 532.6 eV is $H_2O$. The peak at 531.5 eV is attributed to $CO_3^{2-}$ or P=O [26,31–33]. The spectrum of P 2*p* is divided into two peaks, 132.7 and 133.5 eV, which are attributed to $PO_4^{3-}$ and $HPO_4^{2-}$, respectively [34,35]. Al signal is inexistent in the spectrum (Figure 6d), indicating that the XPS signals are mainly attributed to the corrosion products film after immersion test, not from the HT film. The result is in accordance with the above results that there is another film layer on the top of HT film (Figure 5b) and almost no Al is identified in this layer. The high-resolution spectrum of Ca 2*p* displays two distinctive peaks due to the spin orbit splitting. The Ca $2p_{3/2}$ peak at 347.8 eV can be attributed to $CaHPO_4·2H_2O$. The binding energy of Ca $2p_{3/2}$ peak at 346.9 eV can be attributed to $Ca_3(PO_4)_2$ or $CaCO_3$ [34]. The Mg 1*s* peak at 1303.9 eV is assigned to $Mg_3(PO_4)_2$ [35]. It is observed that the content of phosphates is larger than that of hydrogen phosphates. Based on the composition analysis, it can be seen that the top corrosion products layer is mainly consisted of $Mg_3(PO_4)_2$, $Ca_3(PO_4)_2$, $CaHPO_4·2H_2O$ and $CaCO_3$.

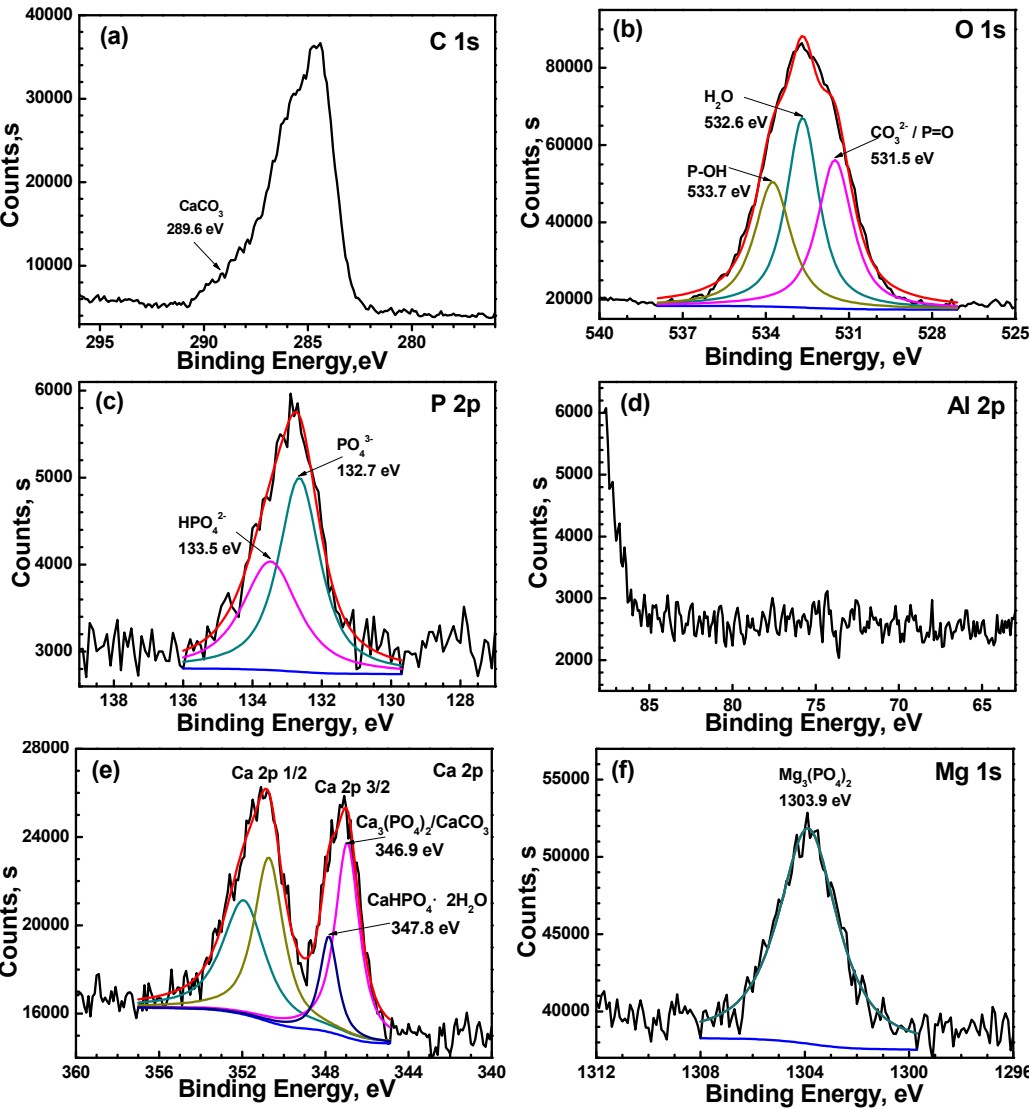

**Figure 6.** XPS analysis of the hydrotalcite film after immersion test for 15 days in Hank's solution: (**a**) C 1*s*; (**b**) O 1*s*; (**c**) P 2*p*; (**d**) Al 2*p*; (**e**) Ca 2*p*; (**f**) Mg 1*s*.

### 3.5. A Comparison of the Corrosion Behavior of AZ31 with HT Film in NaCl Solution and Hank's Solution

We preliminarily compared the corrosion protection effect of the HT film for AZ31 alloy in $0.1$ mol·L$^{-1}$ NaCl and Hank's solution. The corrosion behavior of the HT film in SBF is quite different from that in NaCl solution. In NaCl solution, localized corrosion has already occurred on the coated sample after 12 h immersion (Figure 5e in Reference [26]). When after 3 days immersion, the coated sample displayed a filiform corrosion characteristic and metal brightness (Figure 5a in Reference [28]). As illustrated in Figure 7a, dissolution of the HT film takes place, accordingly, the HT film cannot continuously provide protection to the Mg substrate for longer periods of time, and many long filaments are observed. Differently, in SBF, no corrosion is visible in the region of the coated sample after OCP test for more than 3 days. And, the whole surface of the substrate is still covered with film even after 15 days immersion. However, pit corrosion occurs at some active spots. Although the corrosion products deposited on the pit can provide some degree of protection, the corrosion suppression effect is limited in view of the fact that the products particles deposit is loosely above the pits (Figure 4a). In addition, the electrolyte can penetrate the film at the weak sites during the long-time immersion, and then micro-cracks appear on the surface of the film. Once the electrolyte passes through the micro-cracks to the substrate, underneath corrosion occurs. The corrosion behavior of the HT film in SBF can be illustrated as shown in Figure 7b. The corrosion protective effect of the HT film in SBF is superior to that in NaCl solution. It may be attributed to the corrosion products layer precipitation above the HT film. As a diffusion barrier against electrolyte uptake, the top corrosion products layer governs the dissolution of HT film and suppresses the corrosion activity. These facts indicate that the corrosion environment governs the corrosion behavior of the HT film.

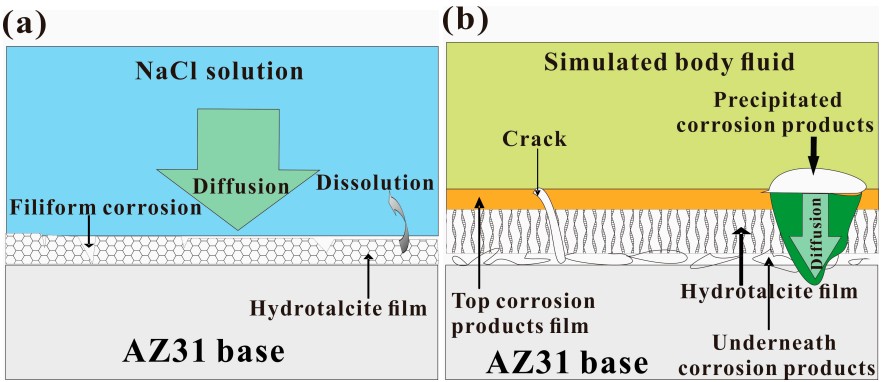

**Figure 7.** Schematic illustration of the corrosion behavior of AZ31 with hydrotalcite film in: (**a**) NaCl solution; (**b**) Hank's solution.

## 4. Summary

In summary, the HT film is compact and uniform, which can improve the corrosion resistance of the AZ31 substrate and greatly delay the initial corrosion in SBF. A dense corrosion products film mainly consisting of Mg/Ca phosphates and CaCO$_3$ is continuously precipitated above the HT film, which has high chemical stability. However, local pit corrosion takes place, and underneath corrosion occurs at some active places after long periods of immersion.

The difference in corrosion behavior of the AZ31 alloy with HT film in SBF and NaCl solution is preliminarily revealed. In NaCl solution, the HT crystals could be dissolved, and macroscopic filiform corrosion occurred only after 3 days immersion. In SBF, corrosion is localized, but most areas of the coated sample still remain integral even after 15 days of immersion. In SBF, the deposited top corrosion products layer can enhance the barrier effect of the HT film. Hence, the HT film can provide protection for longer periods of time in SBF than in NaCl solution.

**Author Contributions:** J.C. and S.M. conceived and designed the experiments. J.C. and K.K. carried out the experimental works and prepared all the figures. J.C. and Y.S. analyzed the data and co-wrote the paper. E.-H.H. and J.A. contributed to the general discussion. All authors reviewed the manuscript.

**Funding:** This work was funded by the National Natural Science Foundation of China (No. 51501156), the Open Research Subject of Key Laboratory of Laboratory of Special Materials and Manufacturing Technology in Sichuan Provincial Universities (No. szjj2016-033), the Research Project of the Education Department of Sichuan Province, China (No. 18ZA0453), the Program of Youth Scientific and Technological Innovation Research Team of Sichuan Province (No. 2019JDTD0024), the Chunhui Program from the Education Ministry of China (No. Z2015094) and the China Postdoctoral Science Foundation Funded Project (No. 2016M602668).

**Conflicts of Interest:** The author declares no conflict of interest.

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
