# Peer review of "Corrosion Behavior of Hydrotalcite Film on AZ31 Alloy in Simulated Body Fluid"

_coatings, doi:10.3390/coatings9020113_

Reviewer 1 Report
This manuscript includes corrosion property of hydrotalicite filme on Mg alloy. Since the corrosion of Mg alloys is a significant issue in actual application fields, this work reporting a new method and corrosion phenomena have a merit to publish. However, there are too many error in this manuscript. So, this manuscript should not be published in current state. The study organized well, but the manuscript didn't professionally write.
These are my critical comments.
biodegradable > the corrosion can be occurred human body. Therefore, the coating shouldn't be biodegradable. Please use appropriate words.
Some incomplete sentences can be found. Ex. Line 120.
Line 150, corrosion current density cna not be changed during potential sweep.
Lin2 153, low current density > ? compared to what? just "low" is not scientific.
What is SEM test? Waht is SEM morphology? please use correct and appropriate terminologies.
"New corrosion product" is wrong. Did you find old corrosion product?
Contact angle can not be a test.
I strongly recommend the author to proof read by native speaker before re-submit.
The contact angle measurements and analysis didn't provide any significant discussioin. In addition, it does not affect the main research flow. Moreover, there are no discussion on the wetting model and surface energy. So, I strongly recommend to remove this part in revised version.
The section of "discussion", it is not discussion. It is summary of previous results. So, the contents are almost identical to previously described analytical results. I couldn't find any new finding or meaning in this part. It means that this part should be removed.
The section of "conlcusion", it is not conclusive. It is summary or the other form of abstract. So, section 5 should be "summary".
Author Response
Thank you for the valuable comments and suggestions. We have modified the manuscript accordingly. The corrections are marked in red in the paper. The responds to the comments are listed below point by point.
biodegradable > the corrosion can be occurred human body. Therefore, the coating shouldn't be biodegradable. Please use appropriate words.
ü This word has been revised. See details in the line 50 in red font.
Some incomplete sentences can be found. Ex. Line 120.
We have revised these sentences carefully. The sentence in line 20 has been removed within the removing of the part of the contact angle measurements. And the other revisions are emphasized in red.
Line 150, corrosion current density cna not be changed during potential sweep.
ü We describe polarization curves, not Potential vs. time curves in Line 150.
Line2 153, low current density > ? compared to what? just "low" is not scientific.
ü This sentence has been revised.
What is SEM test? Waht is SEM morphology? please use correct and appropriate terminologies.
ü Thank you very much for the reviewer’s suggestion. It has been revised.
"New corrosion product" is wrong. Did you find old corrosion product?
ü This formulation has been changed.
Contact angle can not be a test.
ü This part has been removed.
I strongly recommend the author to proof read by native speaker before re-submit.
ü We have revised the English of the manuscript carefully. In addition, we have asked several colleagues who are skilled authors of English language papers to check the English thoroughly.
The contact angle measurements and analysis didn't provide any significant discussioin. In addition, it does not affect the main research flow. Moreover, there are no discussion on the wetting model and surface energy. So, I strongly recommend to remove this part in revised version.
ü This part has been removed.
The section of "discussion", it is not discussion. It is summary of previous results. So, the contents are almost identical to previously described analytical results. I couldn't find any new finding or meaning in this part. It means that this part should be removed.
ü This part has been removed.
The section of "conlcusion", it is not conclusive. It is summary or the other form of abstract. So, section 5 should be "summary".
ü The title of section 4 “Conclusions” has been changed as “Summary”.

Reviewer 2 Report
2.1 Fabrication of the film
Comment 1
HT is prepared using the same procedure in the reference (Corrosion Science 53 (2011) 3281–3288). Authors need to add the paragraph tells the reader that they have followed the same procedure in the Corrosion Science 53 (2011) 3281–3288 for film fabrication.
Comment 2
The method of the film preparation needs to be re-written to be clear for the reader. For example how much the amount of Al is added to make the solution saturated? Regarding the addition of Al panel, it seems that Al dissolved in acidic solution followed by addition of Na2Co3. The sequence needs to be clear for the reader.
Comment 3
Give some information on the AZ31 coupons; i.e., cast? extruded? grain size?
2.2 Characterization
Comment 4
How many times the electrochemical tests were carried out for reproducibility?
3. Results
Comment 5
The delay time of 300 s prior to polarization test is very short. Why the authors choose this short duration of immersion in Hanks’ solution prior to the electrochemical test? The coated and uncoated AZ31 need immersion time longer than 5 min (300 s) to reach the stabilization (where the fluctuation of OCP reaches 10mV per 1000S) prior to the polarization test.
Comment 6
Hydrogen gas generation by rapid corrosion is one of an important issue for Mg alloy as bioimplants. The authors need to comment on whether the coating will simply delay the start of corrosion, or slow down the corrosion speed, and reduce the influence of hydrogen gas?
3.5. Composition analysis of the film
Comment 7
The Authors mentioned that the EDS result shows that the chemical composition of the original HT film only contains C, Mg, Al and O, however, Table 1 shows only Mg, Al and O, please add the C content to the table for region 1 and 2.
4. Discussion
Comment 8
The examination time of 3 days is very short considering clinical applications. The authors need to discuss/comment long term consequence, prediction of the corrosion resistance for a necessary duration.
Author Response
Thank you for the valuable comments and suggestions. We have modified the manuscript accordingly. The corrections are marked in red in the paper. The responds to the comments are listed below point by point.
2.1 Fabrication of the film
Comment 1
HT is prepared using the same procedure in the reference (Corrosion Science 53 (2011) 3281–3288). Authors need to add the paragraph tells the reader that they have followed the same procedure in the Corrosion Science 53 (2011) 3281–3288 for film fabrication.
ü This paragraph has been added. See details in the section 2.1 in red font.
Comment 2
The method of the film preparation needs to be re-written to be clear for the reader. For example how much the amount of Al is added to make the solution saturated? Regarding the addition of Al panel, it seems that Al dissolved in acidic solution followed by addition of Na2Co3. The sequence needs to be clear for the reader.
ü The experimental procedure has been described more clearly and exactly. See details in the section 2.1 in red font. Whereas, the content of Al has not been measured, but which is sufficient to synthetise the Mg-Al HT according to our precious research (J. Chen, Y.W. Song, D.Y. Shan, E.H. Han, Study of the in situ growth mechanism of Mg-Al hydrotalcite conversion film on AZ31 magnesium alloy, Corrosion Science, 2012, 63: 148-158).
Comment 3
Give some information on the AZ31 coupons; i.e., cast? extruded? grain size?
ü The information of the AZ31 coupons has been added. See details in the section 2.1 in red font.
2.2 Characterization
Comment 4
How many times the electrochemical tests were carried out for reproducibility?
ü The electrochemical tests were done at least three times to ensure the correction and repeatability of the data. Only one curve was shown, the date value of which is in the middle.
3. Results
Comment 5
The delay time of 300 s prior to polarization test is very short. Why the authors choose this short duration of immersion in Hanks’ solution prior to the electrochemical test? The coated and uncoated AZ31 need immersion time longer than 5 min (300 s) to reach the stabilization (where the fluctuation of OCP reaches 10mV per 1000S) prior to the polarization test.
ü Yes. The delay time of 300 s prior to polarization test is short. However, you know that a corrosion products film was formed continuously on the surface according to the morphology observation. We worried about that the testing result might not be related to the fresh surface of the samples when the immersion time was prolonged. Thank you for the review’s suggestion. In future research, we will extend the immersion time longer than 5 min in Hanks’ solution prior to the polarization test.
Comment 6
Hydrogen gas generation by rapid corrosion is one of an important issue for Mg alloy as bioimplants. The authors need to comment on whether the coating will simply delay the start of corrosion, or slow down the corrosion speed, and reduce the influence of hydrogen gas?
ü We have mentioned that the coating delayed the start of corrosion and slow down the corrosion speed. See details in the sections of “Electrochemical corrosion test” and “Summary”.
Yes. The hydrogen gas generation rate is very important. The measurement of hydrogen evolution will carried out in the future study.
3.5. Composition analysis of the film
Comment 7
The Authors mentioned that the EDS result shows that the chemical composition of the original HT film only contains C, Mg, Al and O, however, Table 1 shows only Mg, Al and O, please add the C content to the table for region 1 and 2.
ü We have added the C content to Table 1 for region 1 and 2.
4. Discussion
Comment 8
The examination time of 3 days is very short considering clinical applications. The authors need to discuss/comment long term consequence, prediction of the corrosion resistance for a necessary duration.
ü Yes, the examination time of 3 days is short for clinical applications. However, this examination time is for the electrochemical test. When in the immersion test, the examination time was 15 days. And after 15 days immersion, the majority of the coated sample is not attacked. In future research, long term examination will be carried out, especially in the in-vivo test. Thanks very much for the reviewer’s suggestion.

Reviewer 3 Report
The manuscript by Chen et al. is of poor scientific quality. The fundamental problems are the following:
- AZ alloy is tested in simulated body fluid electrolyte. AZ series magnesium alloys contain 3 to 9 % aluminum. The latter has been suspected of causing multiple health problems, including dementia and autism. Aluminum containing magnesium alloys have been excluded from the list of potential candidates for bioresorbable implants. It is pointless to test aluminum containing magnesium alloys in SBF.
- Hydrotacite, HT, Mg6Al2CO3(OH)16·4H2O, used in this work, again, contains Al by default. Why coat bioresorbable implant with harmful aluminum containing layer if multiple hydroxyapatite films, fully biocompatible (as well as MAO/PEO, PLA etc. coatings) have been proven to be successful for this purpose? This is pointless.
- The main message of the manuscript (see abstract and conclusions) is that HT provides better corrosion protection for Mg in SBF compared to NaCl. This is rather useless observation. Does this mean that Mg coated with HT for engineering applications needs to be tested in SBF from now on? NO! Do the authors propose to test Mg aimed for bioapplicaiton in SBF? This has been proven years ago.
- lines 103-104. The authors judge about the chemical composition of the layer formed on Mg simply by looking at SEM picture. This is unacceptable! The authors claimed that microstructure shown in Fig. 1 is HT. Mg(OH)2 looks exactly the same, moreover, there might be tens of other compound that look the same under scanning electron microscope. Moreover, EDS results presented in Table 1 testify for formation of Mg(OH)2 rather than HT. This is no clear evidence that the film characterised is this manuscript is actually the HT.
Considering four main points above, I fail to see the real scientific aim and result of the manuscript in question.
Other (selected) issues:
-The authors did not carry out a rigorous literature survey on the recent advances in understanding degradation of Mg alloys in simulated body fluid electrolytes. Many relevant papers were not cited. Some factual mistakes are repeated in the Intro based on old works and omitting recent findings. For example, it has been recently shown by several groups that pH of SBF electrolytes (HBSS or SBF with no Tris) is not highly alkaline. In contrast to what is stated in the Intro, the pH change is not an issue during in vivo degradation of Mg.
-The problem of too fast degradation of Mg can barely be solved by surface treatment. The main method to solve this problem, development of corrosion resistant alloys, must be outlined in the Intro.
-There are no clear results that actually show better corrosion protection of HT in SBF compared to NaCl. Hydrogen evolution or mass loss results would be convincing, or at least evolution of EIS spectra for 2 to 4 weeks. Presented polarization curves are not conclusive for the purpose because they only show the initial condition (not the evolution in time) and only for bare vs coated alloy. Clearly, coated alloy shows initial slightly better corrosion protection performance. Half of the OCP results is missing, that again makes them inconclusive.
- XPS results are ambiguous. Typically, hydroxyapatite like compounds (partially substituted with Mg and carbonate) are found on Mg surface after exposure in decent SBF electrolytes. XPS obviously, cannot distinguish between hydroxyapatite and calcium phosphate, hydroxyapatite partially substituted with carbonate and calcium carbonate. Thus, the authors end up with ambiguous (if not erroneous) composition of the layers formed after immersion.
Note, as I deem that the manuscript does not deserve to be published, I skipped the analysis of multiple technical and editorial problems, such as missing scale, wrong references, bad English, etc.
Author Response
Thank you for the valuable comments and suggestions. We have modified the manuscript accordingly. The corrections are marked in red in the paper. The responds to the comments are listed below point by point.
- AZ alloy is tested in simulated body fluid electrolyte. AZ series magnesium alloys contain 3 to 9 % aluminum. The latter has been suspected of causing multiple health problems, including dementia and autism. Aluminum containing magnesium alloys have been excluded from the list of potential candidates for bioresorbable implants. It is pointless to test aluminum containing magnesium alloys in SBF.
ü Yes, Al is harmful to health when the content is high. Whereas, AZ31 Mg alloy was also used in other researches. For example, the reference: Laura C. Córdoba, Christophe Hélary, Fátima Montemor, Thibaud Coradin, Bi-layered silane-TiO2/collagen coating to control biodegradation and biointegration of Mg alloys, Materials Science and Engineering: C, 94, 2019, 126-138. Another reference: Changgang Wang, Liping Wu, Fang Xue, Rongyao Ma, Ini-Ibehe Nabuk Etim, Xuehui Hao, Junhua Dong, Wei Ke, Electrochemical noise analysis on the pit corrosion susceptibility of biodegradable AZ31 magnesium alloy in four types of simulated body solutions, Journal of Materials Science & Technology, 2018, 1876-1884.
In addition, the present work is our preliminary work. We want to test the corrosion performance of the HT film as biologic coating in SBF and compare it to the corrosion behavior in NaCl solution. Hence, we also use AZ31 alloy in this work to ensure the same condition with previous work (Chen, J.; Song, Y.W.; Shan, D.Y.; Han, E.H. Modifications of the hydrotalcite film on AZ31 Mg alloy by phytic acid: The effects on morphology, composition and corrosion resistance. Corros. Sci. 2013, 74: 130-138). Now, we are doing some research on the Mg-Fe HT-like film on the MgZn and MgCa series alloys.
- Hydrotacite, HT, Mg6Al2CO3(OH)16·4H2O, used in this work, again, contains Al by default. Why coat bioresorbable implant with harmful aluminum containing layer if multiple hydroxyapatite films, fully biocompatible (as well as MAO/PEO, PLA etc. coatings) have been proven to be successful for this purpose? This is pointless.
The reason for this question is similar to the above problem.
- The main message of the manuscript (see abstract and conclusions) is that HT provides better corrosion protection for Mg in SBF compared to NaCl. This is rather useless observation. Does this mean that Mg coated with HT for engineering applications needs to be tested in SBF from now on? NO! Do the authors propose to test Mg aimed for bioapplicaiton in SBF? This has been proven years ago.
ü No, we propose to test the HT film as a bioactive coating for the Mg alloys. We focused on the influence of the ions in body fluid, such as Ca2+, HCO3− and HPO42−, on the corrosion behavior and corrosion types of the Mg alloy coated with HT film.
- lines 103-104. The authors judge about the chemical composition of the layer formed on Mg simply by looking at SEM picture. This is unacceptable! The authors claimed that microstructure shown in Fig. 1 is HT. Mg(OH)2 looks exactly the same, moreover, there might be tens of other compound that look the same under scanning electron microscope. Moreover, EDS results presented in Table 1 testify for formation of Mg(OH)2 rather than HT. This is no clear evidence that the film characterised is this manuscript is actually the HT.
ü Thank you for the reviewer’s valuable suggestion. The data for chemical composition analysis of the layer were published in our previous work. The sentence telling the reader that the chemical composition of the layer was mainly Mg6Al2(OH)16CO3·4H2O has been supplemented. See details in the section 3.1 in red font. In addition, the EDS results presented in Table 1 have been revised. C was tested in the film (shown as the followed figures), but we did not count its content. The Al signal is also strong. Hence, it can’t be Mg(OH)2.
- The authors did not carry out a rigorous literature survey on the recent advances in understanding degradation of Mg alloys in simulated body fluid electrolytes. Many relevant papers were not cited. Some factual mistakes are repeated in the Intro based on old works and omitting recent findings. For example, it has been recently shown by several groups that pH of SBF electrolytes (HBSS or SBF with no Tris) is not highly alkaline. In contrast to what is stated in the Intro, the pH change is not an issue during in vivo degradation of Mg.
ü These references have been deleted, and the introduction has been revised. See detail in the introduction in red font and references.
- The problem of too fast degradation of Mg can barely be solved by surface treatment. The main method to solve this problem, development of corrosion resistant alloys, must be outlined in the Intro.
ü The introduction has been revised. See detail in the introduction in red font
- There are no clear results that actually show better corrosion protection of HT in SBF compared to NaCl. Hydrogen evolution or mass loss results would be convincing, or at least evolution of EIS spectra for 2 to 4 weeks. Presented polarization curves are not conclusive for the purpose because they only show the initial condition (not the evolution in time) and only for bare vs coated alloy. Clearly, coated alloy shows initial slightly better corrosion protection performance. Half of the OCP results is missing, that again makes them inconclusive.
ü The results show better corrosion protection of HT in SBF compared to NaCl. But, the pictures are not shown in this manuscript, which had been published in previous papers, see details in reference [26] and [28].The comparison was also described in the last paragraph above the summary section. The hydrogen evolution or EIS tests has already been carried out, which will be analyzed in the follow-up work. The OCP result of the bare alloy is missing because it keeps stable. Hence, we stopped the testing.
- XPS results are ambiguous. Typically, hydroxyapatite like compounds (partially substituted with Mg and carbonate) are found on Mg surface after exposure in decent SBF electrolytes. XPS obviously, cannot distinguish between hydroxyapatite and calcium phosphate, hydroxyapatite partially substituted with carbonate and calcium carbonate. Thus, the authors end up with ambiguous (if not erroneous) composition of the layers formed after immersion.
ü Although it is not so convinced to confirm the specific components based on the present EDS and XPS results, XPS results can analysis the composition to a certain extent. In order to determine the detail structure changes of the HT after corrosion in SBF, other test means will be carried out in the future research.

Reviewer 4 Report
Comments are attached.

Author Response
Thank you for the valuable comments and suggestions. We have modified the manuscript accordingly. See details in the PDF file.

Round 2
Reviewer 1 Report
I really appreciate to the author for the point by point revisions. The manuscript is almost completed to be published. However, one critical comments was not corrected.
Previous comment: corrosion current density can not be changed during potential sweep.
Answer: We describe polarization curves, not Potential vs. time curves in Line 150.
I recommend to review basic theories for potentiodynamic polarization curve (e.g., Tafel equation and Butler-Volmer equation). The corrosion current density is a point of intersection of anodic branch and cathodic branch. Therefore, it can not be read directly in the curve and can be estimated from Tafel fitting. For these reason, the claim "the corrosion current density (icorr) increases much more slowly with 137 increasing anodic potential in the passive tendency region compared to that of the curve 1" is truly wrong.
I think the author also does not want to publish with wrong description. Please correct this before publication.
Author Response
Thanks again for the valuable suggestions. We have revised the this sentence. In addition, we also revised the English of the manuscript. See details in the paper in red font.
Reviewer 2 Report
Paper has been improved and is good for publication.
Author Response
Thanks again. We have revised the WHOLE manuscript carefully. Some grammar errors have been revised. See details in the modified manuscript in red font.
Reviewer 3 Report
Minor changes have been introduced to the manuscript instead of requested rejection. Obviously, the main issues were not addressed, because they question the logic on this fragmentary work. Now, the authors actually admit in their answers that they are going to proceed with another Mg alloy and different type of HT-like compounds. This confirms that usage of AZ alloy covered with HT film is senseless for biomedical application. I hold my opinion that this manuscript is of poor quality and should not be published.
Author Response
Thanks again. We know that there is a stomach medicine named Hydrotalcite Tables, which also contains Al. We also revised the English of the manuscript. See details in the paper in red font.